# Differential Levels of Tryptophan–Kynurenine Pathway Metabolites in the Hippocampus, Anterior Temporal Lobe, and Neocortex in an Animal Model of Temporal Lobe Epilepsy

**DOI:** 10.3390/cells11223560

**Published:** 2022-11-10

**Authors:** Soumil Dey, Vivek Dubey, Aparna Banerjee Dixit, Manjari Tripathi, Poodipedi Sarat Chandra, Jyotirmoy Banerjee

**Affiliations:** 1Department of Neurosurgery, All India Institute of Medical Sciences, New Delhi 110029, India; 2Department of Biophysics, All India Institute of Medical Sciences, New Delhi 110029, India; 3Dr. B.R. Ambedkar Centre for Biomedical Research, University of Delhi, Delhi 110007, India; 4Department of Neurology, All India Institute of Medical Sciences, New Delhi 110029, India

**Keywords:** temporal lobe epilepsy, tryptophan–kynurenine pathway, glutamate receptors, kynurenic acid, patch-clamp technique

## Abstract

Glutamate-receptor-mediated hyperexcitability contributes to seizure generation in temporal lobe epilepsy (TLE). Tryptophan–kynurenine pathway (TKP) metabolites regulate glutamate receptor activity under physiological conditions. This study was designed to investigate alterations in the levels of TKP metabolites and the differential regulation of glutamatergic activity by TKP metabolites in the hippocampus, anterior temporal lobe (ATL), and neocortex samples of a lithium–pilocarpine rat model of TLE. We observed that levels of tryptophan were reduced in the hippocampus and ATL samples but unaltered in the neocortex samples. The levels of kynurenic acid were reduced in the hippocampus samples and unaltered in the ATL and neocortex samples of the TLE rats. The levels of kynurenine were unaltered in all three regions of the TLE rats. The magnitude of reduction in these metabolites in all regions was unaltered in the TLE rats. The frequency and amplitude of spontaneous excitatory postsynaptic currents were enhanced in hippocampus ATL samples but not in the neocortex samples of the TLE rats. The exogenous application of kynurenic acid inhibited glutamatergic activity in the slice preparations of all these regions in both the control and the TLE rats. However, the magnitude of reduction in the frequency of kynurenic acid was higher in the hippocampus (18.44 ± 2.6% in control vs. 30.02 ± 1.5 in TLE rats) and ATL (16.31 ± 0.91% in control vs. 29.82 ± 3.08% in TLE rats) samples of the TLE rats. These findings suggest the differential regulation of glutamatergic activity by TKP metabolites in the hippocampus, ATL, and neocortex of TLE rats.

## 1. Introduction

Tryptophan–kynurenine pathway (TKP) metabolites are known to regulate glutamate-receptor-mediated synaptic transmission under physiological conditions. One of the metabolites, kynurenic acid (KYNA), is an endogenous neuromodulator and an inhibitor of all excitatory amino acid receptors at high micromolecular concentrations [1], and it selectively inhibits the glycine co-agonist site of NMDA receptors at low concentrations [2]. KYNA is synthesized by the irreversible conversion of L-kynurenine by kynurenine aminotransferase I and II (KAT I & II), which are found in astrocytes. KAT activity depends on co-factor pyridoxal phosphate (PLP). The tryptophan–kynurenine pathway is the principal catabolic pathway for tryptophan, through which more than 90% of this amino acid is converted to kynurenine by indolamine 2,3-dioxygenase (IDO), the rate-limiting enzyme for this whole pathway [3,4]. Under physiological conditions, glutamate receptor activity in the hippocampal pyramidal neurons is tightly regulated by kynurenic acid [5]. The inhibitory action of kynurenic acid in glutamate receptors is also mediated through presynaptic nicotinic acetyl choline receptors (nAChRs), which may reduce glutamate release [6]. Consequently, alterations in the level of kynurenic acid inside the brain is associated with abnormal synaptic transmission and epileptiform discharges, as evident from experimental models [7,8,9].

Temporal lobe epilepsy (TLE) is a distributed network disorder where the hippocampus and surrounding temporal lobe structures are involved in unprovoked seizure generation, and hippocampal sclerosis is the principal anatomic substrate [10,11,12]. Augmented glutamate receptor activity and the altered expression of NMDA and AMPA receptor subunits contribute to hyperexcitability [12,13,14]. Previously, we have reported that the reduced endogenous synthesis of KYNA, due to altered levels of KAT II and PLP, contribute to the enhanced glutamate receptor activity in hippocampal samples obtained from patients with TLE. However, the exogenous application of kynurenic acid suppressed spontaneous glutamatergic activity in those samples [15]. In the pilocarpine rat model of TLE, we observed differences in the glutamatergic tone between the hippocampus, anterior temporal lobe (ATL), and neocortex, possibly due to the presence of independent epileptogenic networks [14]. The primary aim of this study was to measure the levels of TKP metabolites in the hippocampus, ATL, and prefrontal cortex of TLE rats. In addition, we investigated the effect of exogenously applied KYNA on spontaneous glutamatergic activity in the samples obtained from these three regions.

## 2. Materials and Methods

### 2.1. Animals and Epilepsy Model Development

Adult male Sprague-Dawley rats (SD; 7–8 weeks old, weighing 200–250 g) were kept in groups of 2 per cage in a 12 h light/dark cycle, with water and food *ad libitum*. This study was ethically approved by the Institutional Animal Ethics Committee (File No: 14/IACE-1/2017).

To develop temporal lobe epilepsy, a lithium–pilocarpine model of status epilepticus (SE) was chosen as this model replicates most of the etiopathologies of patients with TLE [16]. The optimal dose of pilocarpine was 250 mg/kg based on our previous study [14]. LiCl (127 mg/kg, i.p.) was injected 24 h before pilocarpine treatment to increase the susceptibility of the brain to pilocarpine. The rats were treated with methyl scopolamine (1 mg/kg, i.p.) 30 min prior to pilocarpine to counteract the peripheral cholinomimetic effects and the lethality of pilocarpine [17]. A total of 60 min after the onset of SE, the convulsions were suppressed by diazepam (10 mg/kg, i.p.). The weight-matched control rats received 0.9% saline solution. The rats were monitored during the initial 15 min for behavioral changes, followed by observation for acute seizures, by well-trained technicians (Figure 1). The rats that showed features of the Racine scale under category IV and V [18] were included in the study.

Following rescue from SE, the rats were euthanized by CO_2_ narcosis and sacrificed by decapitation. Brains were carefully removed and tissues from different regions (anterior portion of hippocampus, middle temporal gyrus for anterior temporal lobe, and prefrontal cortex for frontal neocortex) were collected for various experiments. For in vitro electrophysiology studies, tissues were immediately used for experiments and for metabolites estimation, tissues were stored at −80 °C.

### 2.2. Tissue Preparation for Histology

After rescuing from SE, rats were euthanized by asphyxiation in a CO_2_ chamber, followed by decapitation. Brains were carefully removed from cranial cavities and tissues from different regions (anterior temporal lobe, frontal neocortex, and hippocampus) were collected for experiments. The hippocampus, ATL, and frontal neocortex were fixed in a 0.1 M phosphate buffer containing 4% paraformaldehyde, pH 7.4 for 72 h. After fixation, the tissues were dehydrated, embedded, and cut into 5 μm thick sections with a microtome and collected on poly-L-lysine-coated slides for histological evaluations. Tissue sections were deparaffinized, hydrated, and then rinsed with phosphate-buffered solution (PBS). Nissl staining (cresyl violet staining) was performed as described previously [14,19]. After staining, the sections were dehydrated using ethanol (30%, 50%, 70%, 100%), cleared in xylene, mounted with DPX, and observed under a bright-field upright research microscope (BX53, Olympus).

### 2.3. Estimation of Metabolites

The quantification of the metabolites (tryptophan kynurenine, kynurenic acid, and PLP) was performed by HPLC fluorescence detection. Frozen tissues were thawed, homogenized in an extraction solvent containing 0.1 M formic acid, and centrifuged at 15,000 rpm for 5 min at 0 °C. The supernatant was removed and stored at −80 °C until analysis. The reverse-phase isocratic HPLC was used to separate the metabolites with a Phenomenex C18 column. The column temperature was 37 °C. The mobile phase, pumped at 1 mL/min, consisted of 50 mM acetic acid and 100 mM zinc acetate containing 3% acetonitrile (pH 4.7). The standards (all procured from Sigma Aldrich) were prepared with deionized water. Subsequently, different concentrations were prepared by serial dilution dissolution (0.75, 1.5, 3.15, 6.25, 12.5, 25, 50, 100 ng/mL). A total of 10 µL of a standard and sample volume was injected into the system through an autosampler. The detection system was a fluorescence detector with a dual-wavelength simultaneous-monitoring capability (Shimadzu). The retention time for tryptophan was 21 min, 36 min for kynurenine, 35 min for kynurenic acid, and 6.5 min for pyridoxal phosphate. The excitation and emission wavelength spectra of tryptophan, kynurenine, kynurenic acid, and PLP were 297/348, 365/480, 344/404 and 300/400 nm, respectively [15,20,21].

The activity for IDO and KAT II was calculated from ratio between products and substrates (kynurenine–tryptophan for IDO and kynurenic acid–kynurenine for KAT II).

### 2.4. In Vitro Electrophysiology

Immediately after decapitation, the brains were kept in ice-cold artificial cerebrospinal fluid (ACSF) composed of NaCl, 125 mM; KCl, 2 mM; CaCl_2_, 2 mM; NaHCO_3_, 25 mM; NaH_2_PO_4_, 1.25 mM; MgCl_2_; and 1 mM Glucose, 25 mM. The ACSF was bubbled with 95% O_2_ and 5% CO_2_. The regions of interest from the hippocampi, anterior temporal lobes (ATL), and of the frontal neocortices were dissected out and mounted on the stage of a vibratome (VT1000S, Leica), which was used to cut transversal slices of 350–400 μm thickness. The slices were stored at room temperature for at least 45 min in an immersion chamber containing ACSF continuously bubbled with 95% O_2_ and 5% CO_2_ before recordings [13,14,15]. Pyramidal neurons with thick soma and a single tapering dendrite were visually identified using IR–DIC video-microscopy. Patch pipettes with a resistance of 3–5 MΩ were filled with an internal solution containing HEPES, 10 mM; MgCl_2_, 2 mM, Cs-methanesulfonate, 130 mM; EGTA, 10 mM; and CsCl, 10 mM. Spontaneous excitatory postsynaptic currents (sEPSCs) were recorded under the whole-cell configuration of the patch-clamp technique at a holding potential of −70 mV from the pyramidal neurons using amplifier Axopatch 200B (Molecular Devices, San Jose, CA, USA). To record the sEPSCs in the presence of exogenously applied KYNA, the slices were perfused with ACSF containing 10 μM KYNA for 30 min, and then the spontaneous activities were recorded. Data were analyzed in PCLAMP 10.0 software (Molecular Devices, San Jose, CA, USA). The frequency, amplitude, rise time (10–90%), and decay time constant (τ_d_) of the synaptic events were measured. All recordings were visually inspected to select events which showed a steep rising and an exponential decay phase for the kinetic analysis of EPSCs. Events that showed multiple peaks were excluded for kinetic analysis but included for frequency calculation as multiple events. We recorded spontaneous EPSCs from one neuron per slice preparation obtained from each rat. Hence, the number of samples/neurons represent number of slice preparations per rat.

### 2.5. Statistical Analysis

Statistical analyses were performed on GraphPad Prism 8.0 software (version 8.0.1). The data were presented as mean ± SEM. Analysis between two groups were performed by Mann–Whitney tests. *p* < 0.05 was considered significant.

## 3. Results

### 3.1. Histopathological Features of Pilocarpine (TLE) Treated Rats

A total of 90 min after administration of pilocarpine, brain tissue was collected for examination. CV staining was performed to study the histopathological changes in the TLE rat model of epilepsy in a region-specific manner. A disrupted and altered layer organization was observed in the hippocampus of TLE rats compared to the controls (Figure 2A,B). Similar morphological changes were observed in the ATL samples of the TLE rats compared to the controls (Figure 2C,D). Neuronal damages were also observed in the hippocampus and ATL as evidenced by the pyknotic nucleus. No noticeable changes were observed in the neo-cortical samples of the TLE rats compared to the controls (Figure 2E,F).

### 3.2. The Levels of Tryptophan–Kynurenine Pathway Metabolites and Enzyme Activities Were Altered in Acute Model of TLE

We quantified tryptophan, kynurenine, and kynurenic acid in the hippocampal, ATL, and frontal neocortex samples from both the control and TLE rats. The level of tryptophan was significantly reduced in the hippocampal (2996 ± 2017 ng/μg of tissue vs. 24,318 ± 10,056 ng/μg of tissue) and ATL (7769 ± 2997 ng/μg of tissue vs. 48,449 ± 14,074 ng/μg of tissue) samples of the TLE rats compared to that of the respective control rats (Figure 3A). The level of kynurenine was unchanged in the hippocampal (6567 ± 2311 ng/μg of tissue vs. 2182 ± 668 ng/μg of tissue) and ATL (10,165 ± 6556 ng/μg of tissue vs. 2295 ± 871.2 ng/μg of tissue) samples between the two groups (Figure 3B). The level of kynurenic acid was significantly reduced in the hippocampal samples of the TLE rats compared to that of the respective control rats (62.7 ± 14.46 ng/μg of tissue vs. 149 ± 47.82 ng/μg of tissue) (Figure 3C), but that of the ATL was unaltered (156.1 ± 61.89 ng/μg of tissue vs. 150.8 ± 41.04 ng/μg of tissue). The level of these metabolites was unaltered in the frontal neocortex samples between the two groups (for tryptophan, 8341 ± 4281 ng/μg of tissue vs. 18,487 ± 6439 ng/μg of tissue; for kynurenine, 745.7 ± 319.6 ng/μg of tissue vs. 1559 ± 487.1 ng/μg of tissue; for kynurenic acid, 25.75 ± 5.891 ng/μg of tissue vs. 87.93 ± 32.4 ng/μg of tissue). We did not observe any significant change in the magnitude of alteration in the levels of tryptophan, kynurenine, and kynurenic acid in the hippocampus, ATL, and neocortex of TLE rats with respect to the controls.

The functional activities of IDO and KAT II were expressed in terms of the ratio between product: substrate of those enzymes. The ratio of kynurenine–tryptophan was significantly reduced in the hippocampal and ATL samples of the TLE rats compared to that of the respective control rats, implying a higher IDO activity in those samples (Figure 3D). The ratio of kynurenic acid–kynurenine was significantly elevated in the hippocampal samples of the TLE rats compared to that of the respective control rats, implying a reduced KAT II activity in those samples (Figure 3E). The level of PLP, a co-factor of KAT II, was unaltered between the two groups (Figure 3F).

### 3.3. Spontaneous Glutamatergic Activity Was Higher in the Hippocampus and ATL but Not in Frontal Neocortex in Acute Model of TLE

The spontaneous EPSCs were recorded from CA1 pyramidal neurons of the hippocampus and pyramidal neurons of the ATL and frontal neocortex from both the control and TLE rats at a −70 mV holding potential. These currents were abolished following the perfusion of these slices with an admixture of 50 µM APV (NMDA receptor inhibitor) and 10 µM CNQX (AMPA receptor inhibitor), implying that these were mediated by glutamate receptors. The frequency and amplitude of these events were significantly higher in the hippocampal and ATL samples of the TLE rats compared to that of the respective control rats (Table 1), the but rise time and τ_d_ of these events were not affected. In the neocortex of the control and TLE rats, the above-mentioned parameters of these events were not altered (Table 1).

### 3.4. Exogenously Applied Kynurenic Acid Suppressed the Spontaneous Glutamatergic Activity in Acute Model of TLE

To observe the effect of exogenously applied kynurenic acid on spontaneous glutamatergic activity, the slices were perfused with 10 μM kynurenic acid for 30 min. After 30 min, when the bath solution was completely saturated with kynurenic acid, the spontaneous EPSCs were then recorded. The frequency and amplitude of the spontaneous events following 30 min perfusion were significantly reduced in all regions of both groups. However, the percentage of reduction in frequency was significantly higher in the hippocampal and ATL samples following perfusion with kynurenic acid in the TLE rats compared to that of the control rats (18.44 ± 2.5% in control hippocampus vs. 30.02 ± 1.5% in TLE hippocampus; 16.31 ± 0.9% in control ATL vs. 29.82 ± 3% in TLE ATL; 20.65 ± 2.6% in control neocortex vs. 19.55 ± 3.4% in TLE neocortex; Figure 4B). The percentage of reduction in the amplitude was significantly higher only in the ATL samples following perfusion with kynurenic acid in the TLE rats compared to that of the control rats (20.18 ± 3.1% in control hippocampus vs. 24.83 ± 2.06% in TLE hippocampus; 18.24 ± 0.88% in control ATL vs. 24.73 ± 1.07% in TLE ATL; 19.41 ± 2.4% in control neocortex vs. 22.04 ± 2.6% in TLE neocortex; Figure 4C). The rise time and τ_d_ of these events were not affected.

## 4. Discussion

In the present study we demonstrated that (i) levels of tryptophan were reduced in the hippocampal and ATL samples of the TLE rats, possibly due to the enhanced activity of IDO; (ii) levels of kynurenic acid were reduced in the hippocampal samples of the TLE rats, possibly due to the reduced activity of KAT II; (iii) exogenously applied kynurenic acid suppressed the glutamate-receptor-mediated hyperexcitability in the hippocampal and ATL samples of the TLE rats and the magnitude of suppression was higher than that of the control rats; and (iv) no alterations in the levels of these metabolites and glutamate receptor activity were observed in the neocortical samples of the TLE rats.

Astrocyte-derived kynurenic acid regulates glutamate-receptor-mediated excitatory transmission under physiological conditions. Therefore, it is logical to assume that the alteration in the levels of kynurenic acid disrupt glutamate receptor activity. We reported earlier that the reduced synthesis of endogenous kynurenic acid was due to the reduction in both KAT II expression and its cofactor PLP in patients with TLE [15]. Here, we found reduced activity of the KAT II and a decreased kynurenic acid–kynurenine ratio in the hippocampal samples of the TLE rats, which is in line with our previous study [15]. However, the level of PLP remained unchanged in the hippocampal samples of TLE rats. Although the glutamate receptor activity was higher than that of the control rats, we did not observe an alteration in the levels of kynurenic acid in the ATL samples of the TLE rats. This could be due to the difference in the mechanism of hyperexcitability in the hippocampus and ATL, as reported in patients with TLE [13]. The reduction in the levels of tryptophan observed in the TLE rats could be due to the enhanced activity of IDO (kynurenine–tryptophan ratio) in the hippocampal samples. The increased IDO activity is an Indication of an abnormal tryptophan–kynurenine pathway and an imbalance between kynurenic acid and quinolinic acid, documented in various neurological disorders [3]. As the level of KYNA was reduced following pilocarpine administration, it is possible that KYN was metabolized through the long arm of the tryptophan–kynurenine pathway converting to 3-hydroxykynurenine and finally to quinolinic acid (QUIN). QUIN, being a neurotoxic metabolite, further enhanced and aggravated the NMDA-receptor-mediated hyperexcitability. As the level of KYNA was reduced, it is possible that the increased IDO-activity-dependent elevation of the level of QUIN might be aggravated by the effect of pilocarpine. In addition to the acute phase, IDO activity was also enhanced over a chronic period, followed by an animal model of status epilepticus, as well as in epilepsy patients [22]. Further studies involving the estimation of tryptophan–kynurenine pathway metabolites, including PLP, over a chronic period will provide more insight into this abnormal tryptophan–kynurenine pathway metabolism after TLE.

The increase in the frequency of spontaneous EPSCs observed in the case of the hippocampal and ATL samples may be due to an enhanced presynaptic α7-nAChR-dependent multi-button glutamate release, as well as action-potential-dependent presynaptic excitatory inputs [6,13,23]. The increase in postsynaptic NMDA and AMPA receptor densities may contribute to the increase in the amplitude of spontaneous EPSCs [13,14,24,25]. As kynurenic acid regulates glutamate release through α7 nAChRs [6], it is possible that exogenously applied kynurenic acid suppressed the presynaptic action-potential-dependent glutamate release, causing a reduction in frequency through binding to postsynaptic glutamate receptors, causing a reduction in the amplitude of spontaneous EPSCs. The enhanced α7-nAChR-mediated glutamate release [22] and postsynaptic receptor density may be responsible for the increase in the percentage change of the frequency and amplitude in the hippocampal and ATL samples of the TLE rats.

Here, we did not observe any alteration in kynurenine pathway metabolites and glutamate receptor activity in the frontal neocortex samples of TLE rats. It is possible that, in an acute model of TLE, the epileptogenic zone did not extend up to the frontal neocortex but rather it was restricted to the temporal lobe region. TLE-induced enhancement in the levels of glutamate and GABA are reported to be limited to the hippocampus [26]. This could be the possible reason for the absence of any significant alteration in glutamate receptor activity in the frontal cortex of TLE rats. Experiments in animal models with a chronic period after pilocarpine administration might show the involvement of the frontal neocortex as well.

Understanding the role of kynurenic acid in synaptic transmission in this preclinical epilepsy model has provided a significant amount of information about the disease because the regulation of the glutamate receptor function in several areas of the brain is essential for the prevention of the chronic, unprovoked, recurrent seizure activity. Here, we have shown the potency of kynurenic acid to suppress hyperexcitability, suggesting the potential of kynurenic acid as a therapeutic tool for the regulation of hyperactive glutamate receptors in patients with temporal lobe epilepsy. The influence of diazepam on tryptophan–kynurenine pathway metabolites cannot be ruled out. However, we could not check this in the present study. Further, we cannot rule out the possible diazepam-mediated modulation of glutamatergic activity.

In conclusion, this study shows a tight association between the magnitude of change in the levels of TKP metabolites with the enhanced glutamatergic tone in the hippocampal, ATL, and neocortical samples of TLE rats. Further, we observed the differential regulation of glutamatergic activity by KYNA in samples obtained from these three regions in TLE rats. These findings will further help validate the conjecture that in TLE glutamatergic synaptic activity to form an epileptogenic network in these three regions could be independent of each other.

## Figures and Tables

**Figure 1 cells-11-03560-f001:**
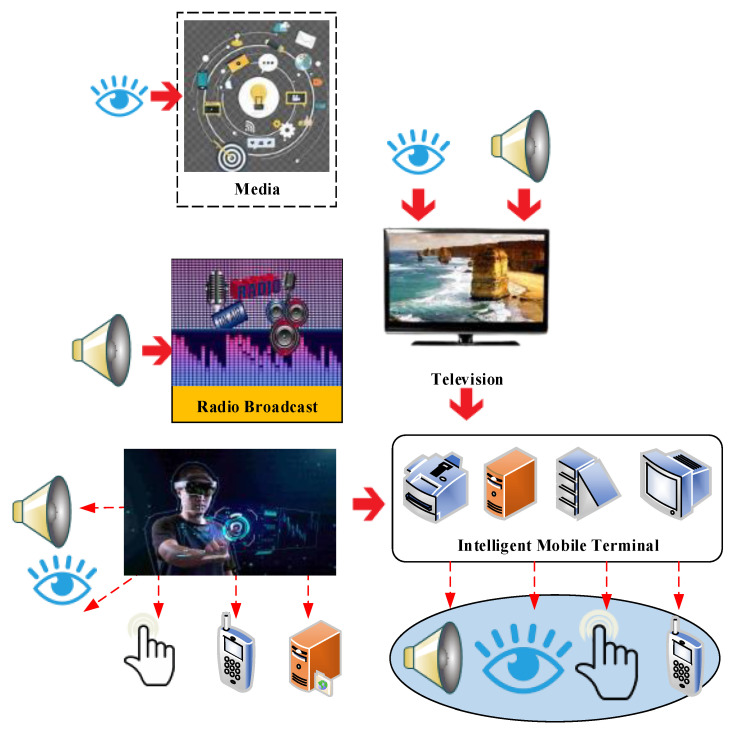
Timescale for creation of lithium–pilocarpine model of status epilepticus. (B.W.: body weight).

**Figure 2 cells-11-03560-f002:**
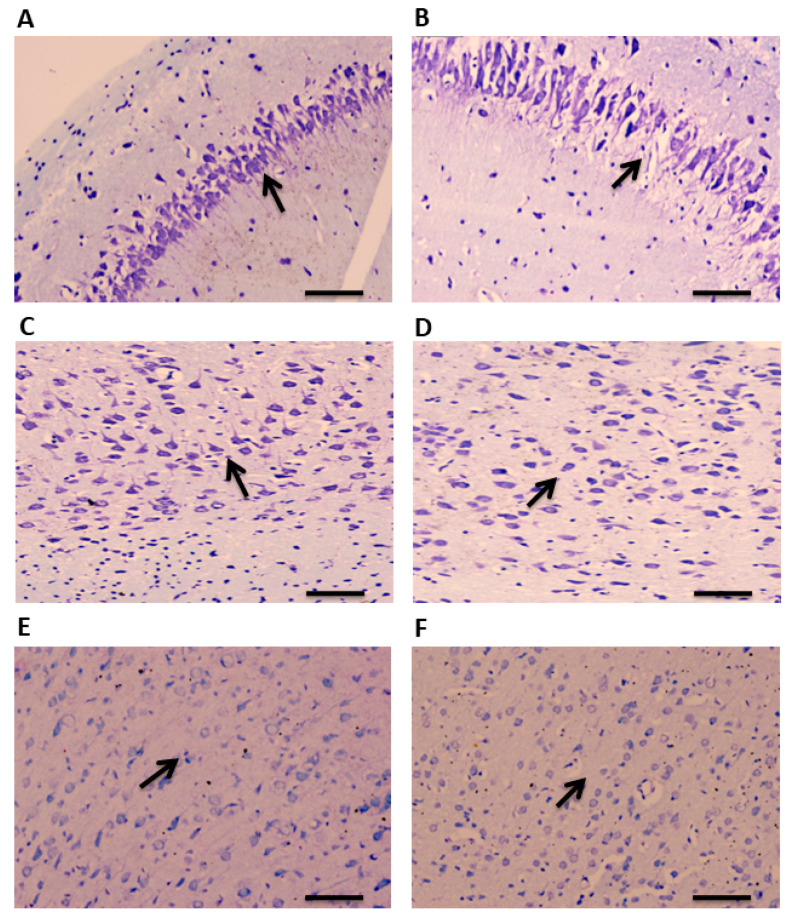
CV staining showing cellular organization in the hippocampus, ATL, and neocortex of TLE rats. Arrows indicates morphological changes and disrupted organization of layers in the hippocampal samples of TLE rats compared to control rats (**A**,**B**). Arrows show altered morphological changes and neuronal organization in the ATL samples of TLE rats compared to control rats (**C**,**D**). No significant changes in the morphology and neuronal organization were observed in the neocortical samples of TLE rats compared to control rats (**E**,**F**). Scale bars: 20 μm.

**Figure 3 cells-11-03560-f003:**
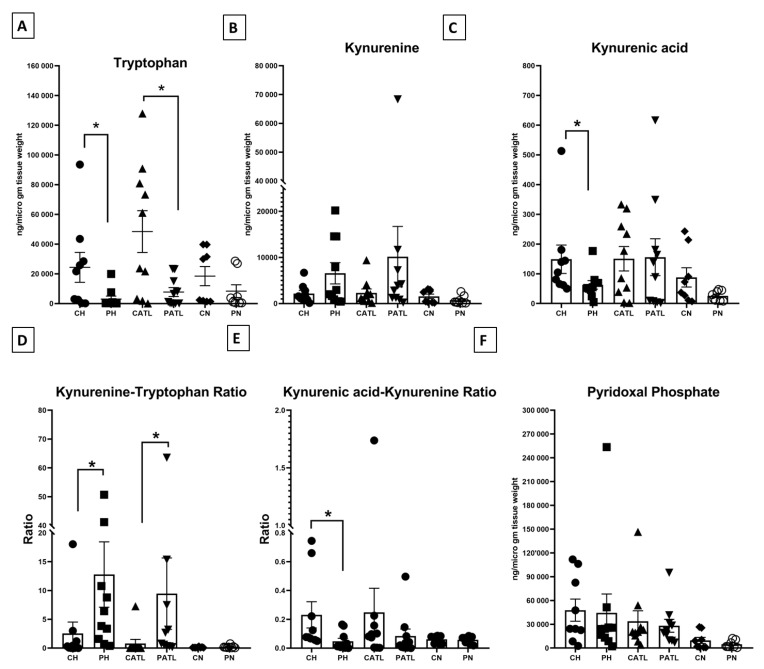
Quantitative estimation of tryptophan–kynurenine pathway metabolites. (**A**) Concentration of tryptophan, (**B**) concentration of kynurenine, (**C**) concentration of kynurenic acid, (**D**) kynurenine–tryptophan ratio, (**E**) kynurenic acid–kynurenine ratio, (**F**) concentration of pyridoxal phosphate. CH, control hippocampus (n = 9); PH, pilocarpine hippocampus (n = 10); CATL, control ATL (n = 10); PATL, pilocarpine ATL (n = 10); CN, control neocortex (n = 8); PN, pilocarpine neocortex (n = 8). Data represented as mean ± SEM; * *p* < 0.05, Mann–Whitney U test.

**Figure 4 cells-11-03560-f004:**
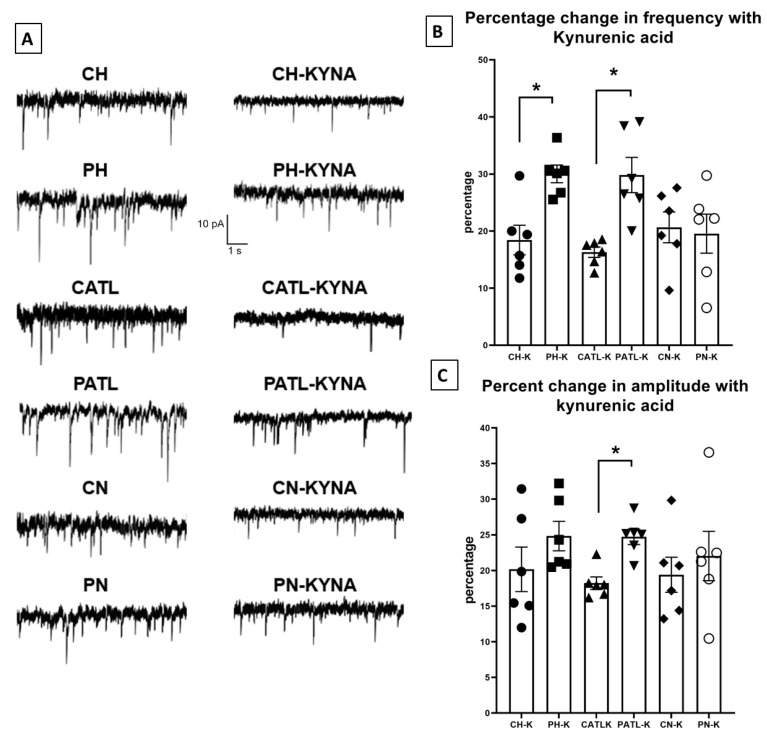
Spontaneous excitatory postsynaptic currents were suppressed after 30 min of perfusion with 10 μM kynurenic acid in the hippocampal and ATL samples of the TLE rats. (**A**) Representative traces in control condition and after perfusion with kynurenic acid from different groups. (**B**,**C**) Percentage reduction in frequency and amplitude of the spontaneous excitatory postsynaptic currents after 30 min perfusion with kynurenic acid. CH, control hippocampus; PH, pilocarpine hippocampus; CATL, control ATL; PATL, pilocarpine ATL; CN, control neocortex; PN, pilocarpine neocortex; KYNA, kynurenic acid. Control rat, n = 10; TLE rat, n = 10; Data represented as mean ± SEM; * *p* < 0.05, Mann–Whitney U test.

**Table 1 cells-11-03560-t001:** Characteristics of EPSCs (frequency, amplitude, rise time, and decay time constant) recorded from pyramidal neurons in hippocampal, ATL, and neocortical samples before and after 30 min perfusion with 10 micro molar kynurenic acid. ‘a’ significant difference with respect to control hippocampus, ‘b’ significant difference with respect to pilocarpine hippocampus, ‘c’ significant difference with respect to control ATL, ‘d’ significant difference with respect to pilocarpine ATL, ‘e’ significant difference with respect to control neocortex, ‘f’ significant difference with respect to pilocarpine neocortex. Data represented as mean ± SEM; Single digit denotes *p* < 0.05. Mann–Whitney U test.

**Parameters**	**Control Hippocampus (10)**	**Control Hippocampus-KYNA (10)**
Frequency (Hz)	0.68 ± 0.04	0.55 ± 0.07 ^a^
Amplitude (pA)	12.05 ± 0.16	10.02 ± 0.38 ^a^
Rise time (ms)	1.9 ± 0.7	2.0 ± 0.6
Decay time constant (τ_d_, ms)	10.6 ± 1.2	9.9 ± 1.7
	**Pilocarpine Hippocampus (10)**	**Pilocarpine Hippocampus-KYNA (10)**
Frequency (Hz)	0.98 ± 0.04 ^a^	0.68 ± 0.05 ^b^
Amplitude (pA)	14.70 ± 0.92 ^a^	11.06 ± 1.13 ^b^
Rise time (ms)	2.0 ± 0.9	2.8 ± 0.5
Decay time constant (τ_d_, ms)	11.4 ± 2.2	10.9 ± 0.8
	**Control ATL (10)**	**Control ATL-KYNA (10)**
Frequency (Hz)	0.67 ± 0.04	0.56 ± 0.03 ^c^
Amplitude (pA)	12.51 ± 0.6	10.23 ± 0.62 ^c^
Rise time (ms)	3.1 ± 0.3	2.9 ± 0.5
Decay time constant (τ_d_, ms)	12.3 ± 3.2	11.5 ± 2.8
	**Pilocarpine ATL (10)**	**Pilocarpine ATL-KYNA (10)**
Frequency (Hz)	1.10 ± 0.09 ^c^	0.77 ± 0.05 ^d^
Amplitude (pA)	16.16 ± 1.04 ^c^	12.17 ± 0.99 ^d^
Rise time (ms)	2.7 ± 0.3	2.3 ± 0.9
Decay time constant (τ_d_, ms)	13.5 ± 2.6	12.9 ± 3.1
	**Control Neocortex (10)**	**Control Neocortex-KYNA (10)**
Frequency (Hz)	0.70 ± 0.16	0.54 ± 0.02 ^e^
Amplitude (pA)	12.53 ± 1.15	10.13 ± 1.5 ^e^
Rise time (ms)	2.9 ± 0.7	2.7 ± 0.4
Decay time constant (τ_d_, ms)	11.7 ± 3.4	11.5 ± 2.0
	**Pilocarpine Neocortex (10)**	**Pilocarpine Neocortex KYNA (10)**
Frequency (Hz)	0.74 ± 0.12	0.52 ± 0.06 ^f^
Amplitude (pA)	12.17 ± 0.41	9.5 ± 1.17 ^f^
Rise time (ms)	2.7 ± 0.6	2.3 ± 0.9
Decay time constant (τ_d_, ms)	12.6 ± 0.8	11.4 ± 0.9

## Data Availability

Not applicable.

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
