# Peer review of "Differential Levels of Tryptophan–Kynurenine Pathway Metabolites in the Hippocampus, Anterior Temporal Lobe, and Neocortex in an Animal Model of Temporal Lobe Epilepsy"

_cells, 2022, doi:10.3390/cells11223560_

Round 1
Reviewer 1 Report
Manuscript entitled „Differential levels of tryptophan-kynurenine pathway metabolites in the hippocampus, anterior temporal lobe and neocortex in temporal lobe epilepsy” by Dey et al. is aimed to investigate the effect of status epilepticus on kynurenine branch of tryptophan metabolism in brain tissue. The idea is sound but the manuscript needs to be revised.
· title - add „in an animal model of temporal lobe epilepsy”
· abstract – add “lithium-pilocarpine rat model of TLE”. It is necessary because lithium persists in cells for a long-period of time and can affect intracellular processes.
· after the first entry of an abbreviation, use only the abbreviation in the rest of the manuscript
· Methods - describe in more detail:
(a) how long the animals were observed for occurrence of convulsions?
(b) after what time diazepam was administered? What does it mean “repeated as needed”? Could this affect the in vitro electrophysiology results?
(c) what does it mean “for spontaneous seizures”? Spontaneous seizures do not appear until 2-3 weeks after pilocarpine-induced status epilepticus. Please explain what was really evaluated?
(d) based on information: “following rescue from SE….” and “ brains were carefully removed…” it can be assumed that all following experiments are related to the acute seizures induced by pilocarpine including SE and not the spontaneous convulsion period. This should be corrected throughout the manuscript.
(e) it would be most appropriate to present a relevant graphic with a time scale
(f) since the HPLC method of detection of tryptophan metabolites utilized in this study was never used to measure such metabolites in brain tissue please provide appropriate data on sensitivity of the method, specify limit of detection of determined substances.
· Statistics – in the study the mean ± SEM were presented however Mann-Whitney test was used to analyze statistical significance; median value would be more appropriate. In addition, the SEM value given in the results is very large, which indicates a large divergence of results. It is alarming that results which differ by 2 and 3 times are not statistically significant (see line 168-171). In this situation it would be recommended to show all individual results (scatter) on the graph, use estimation statistics and recalculate statistical significance. Showing different levels of significance (p<0.05, p<0.01, p<0.001) is not justified. Just specify "at least p<0.05".
· Results, histopathological features - It is critical to know at how long after the administration of PILO the brain tissue was collected for examination. Was it possible for neuronal damage to occur? The authors did not explain what the arrows in Figure1 indicate.
· kynurenine to kynurenic acid ratio does not represent KAT II activity. In fact, KAT II activity was not determined in the study. This should be corrected throughout the manuscript.
· Chapter 3.2 and Fig. 2 – why IDO and KAT activity was defined as substrate to product ratio? Usually the opposite is true, IDO activity is defined as the ratio of KYN to TRP, so is KAT activity (kynurenic acid to kynurenine ratio). The proposed method of calculating enzyme activity will make it easier to understand the results. Just try to explain an increase in enzyme activity (Fig. 2E, PH vs CH) and simultaneous decrease of product content (Fig. 2C, PH vs CH).
· Chapter 3.4 and Fig. 3 – values presented in Fig. 3 does not correspond to that described in text. Please, reevaluate carefully your results. Explain how percentage of reduction was calculated? Clearly state if the larger reduction means that KYNA has a stronger effect in the tissue exposed to pilocarpine? How can you explain this observation? Describe your opinion on this matter in the discussion.
· Line 284-285 “manipulated in several ways” – describe in more detail or provide at least a bibliography.
· There are a lot of spelling errors throughout the manuscript and in the figures.
Author Response
Reviewer 1
Comment: Manuscript entitled „Differential levels of tryptophan-kynurenine pathway metabolites in the hippocampus, anterior temporal lobe and neocortex in temporal lobe epilepsy” by Dey et al. is aimed to investigate the effect of status epilepticus on kynurenine branch of tryptophan metabolism in brain tissue. The idea is sound, but the manuscript needs to be revised.
Response: We thank the reviewer for going through our manuscript and providing valuable comments on our manuscript. We have now provided responses to all the queries raised by the reviewer and most of them has been incorporated in the revised manuscript.
Comment: title - add „in an animal model of temporal lobe epilepsy”
Response: We have added the same and highlighted in the revised manuscript (Line 4).
Comment: abstract – add “lithium-pilocarpine rat model of TLE”. It is necessary because lithium persists in cells for a long-period of time and can affect intracellular processes.
After the first entry of an abbreviation, use only the abbreviation in the rest of the manuscript
Response: We have added the same and highlighted in the revised manuscript (Line 16).
Methods - describe in more detail:
Comment: how long the animals were observed for occurrence of convulsions?
Response: The animals were observed for a total 90 min following pilocarpine injection for occurrence of convulsions.
Comment: after what time diazepam was administered? What does it mean “repeated as needed”? Could this affect the in vitroelectrophysiology results?
Response: Diazepam was administered 60 min after pilocarpine injection when status epilepticus was achieved.
We have removed the phrase “repeated as needed” from the manuscript. We have included in this study only those rats who received a single dose of diazepam. We have recorded here the glutamate receptor activity followed by status epilepticus which is not under regulation of diazepam. It has been reported that 24 h post-injection with diazepam, there was increase in the ratio between AMPA and NMDA receptor-mediated excitatory currents using whole-cell patch-clamp configuration in mouse ventral tegmental area slice preparations (Heikkinen et al., Neuropsychopharmacology, 34:290–298, 2009). In our study we can not rule out the possible diazepam-mediated modulation of glutamatergic activity recorded from the hippocampus of TLE rats. We have now mentioned this as a limitation of our study in the discussion section of the revised manuscript (Line 311).
Comment: what does it mean “for spontaneous seizures”? Spontaneous seizures do not appear until 2-3 weeks after pilocarpine-induced status epilepticus. Please explain what was really evaluated.
Response: We thank the reviewer for pointing out this mistake. We have now removed “for spontaneous seizures” from the manuscript. The present study was focused on the acute changes followed by status epilepticus induction. We have measured the spontaneous glutamatergic activity from pyramidal neurons in the hippocampal samples of TLE rats.
Comment: based on information: “following rescue from SE….” and “brains were carefully removed…” it can be assumed that all following experiments are related to the acute seizures induced by pilocarpine including SE and not the spontaneous convulsion period. This should be corrected throughout the manuscript.
Response: We have now removed the mention of “spontaneous seizure” throughout the manuscript. We agree with the reviewer that in this study we have investigated the acute seizures induced by pilocarpine and not the spontaneous seizures.
Comment: it would be most appropriate to present a relevant graphic with a time scale
Response: We have now added a time scale for status epilepticus induction in the revised manuscript as figure 1.
Comment: since the HPLC method of detection of tryptophan metabolites utilized in this study was never used to measure such metabolites in brain tissue please provide appropriate data on sensitivity of the method, specify limit of detection of determined substances.
Response: One of our recent publications has the exact same HPLC method for quantification of tryptophan metabolites from human brain samples. Here we have replicated the same HPLC method.
We have added sufficient technical details in the HPLC method part and added our previous reference (Line 109-111, 113-115).
Comment: Statistics – in the study the mean ± SEM were presented however Mann-Whitney test was used to analyze statistical significance; median value would be more appropriate. In addition, the SEM value given in the results is very large, which indicates a large divergence of results. It is alarming that results which differ by 2 and 3 times are not statistically significant (see line 168-171). In this situation it would be recommended to show all individual results (scatter) on the graph, use estimation statistics and recalculate statistical significance. Showing different levels of significance (p<0.05, p<0.01, p<0.001) is not justified. Just specify "at least p<0.05".
Response: We have now added scatter plot images of all the results showing individual values of each parameter. We have also mentioned only p<0.05 in the result of quantification of the metabolites (Line 201).
Comment: Results, histopathological features - It is critical to know at how long after the administration of PILO the brain tissue was collected for examination. Was it possible for neuronal damage to occur? The authors did not explain what the arrows in Figure1 indicate.
Response: 90 min after administration of pilocarpine, brain tissue was collected for examination. The arrows indicate the neuronal layers damaged and changed morphology. We have revised the manuscript and figure 2 legend to reflect this (Line 161-165)
Comment: kynurenine to kynurenic acid ratio does not represent KAT II activity. In fact, KAT II activity was not determined in the study. This should be corrected throughout the manuscript.
Response: We have now corrected this throughout the manuscript.
Comment: Chapter 3.2 and Fig. 2 – why IDO and KAT activity was defined as substrate to product ratio? Usually the opposite is true, IDO activity is defined as the ratio of KYN to TRP, so is KAT activity (kynurenic acid to kynurenine ratio). The proposed method of calculating enzyme activity will make it easier to understand the results. Just try to explain an increase in enzyme activity (Fig. 2E, PH vs CH) and simultaneous decrease of product content (Fig. 2C, PH vs CH).
Response: According to the reviewer’s suggestion we have now calculated the IDO and KAT activity by product to substrate ratio (KYN:TRP for IDO and KYNA:KYN for KATII) and modified the figures 3D, 3E as well as in the manuscript (Line 119-120).
Comment: Chapter 3.4 and Fig. 3 – values presented in Fig. 3 does not correspond to that described in text. Please, reevaluate carefully your results. Explain how percentage of reduction was calculated? Clearly state if the larger reduction means that KYNA has a stronger effect in the tissue exposed to pilocarpine? How can you explain this observation? Describe your opinion on this matter in the discussion.
Response: We thank the reviewer for pointing out the discrepancy. We have now mentioned the correct values of percentage change in the revised manuscript (Line 231-233, 235-237)
Comment: Line 284-285 “manipulated in several ways” – describe in more detail or provide at least a bibliography.
Response: We have removed this sentence from the revised manuscript.
Comment: There are a lot of spelling errors throughout the manuscript and in the figures.
Response: We have now corrected the grammatical errors throughout the manuscript.

Reviewer 2 Report
The authors conducted a study to measure the levels of metabolites of the kynurenine pathway in several region of the brain and their relationship with glutaminergic activity regulation in rats with temporal lobe epilepsy. This is a well-written study with good methodology and clearly represented results. The authors should be commended for their work and contribution to better understanding of role of kynurenine pathway and epilepsy pathogenesis and treatment. Please find attached my comment below:
1. Page 10, line 255: you briefly mentioned the role of abnormal activity in various neurological disorders. Recently, a study by Deng et al. (PMID: 33679331) evaluated the application of IDO1 and different kynurenine pathway metabolites in treatment of epilepsy. This reference would be a useful asset to your manuscript.
Author Response
Comment: The authors conducted a study to measure the levels of metabolites of the kynurenine pathway in several region of the brain and their relationship with glutaminergic activity regulation in rats with temporal lobe epilepsy. This is a well-written study with good methodology and clearly represented results. The authors should be commended for their work and contribution to better understanding of role of kynurenine pathway and epilepsy pathogenesis and treatment. Please find attached my comment below: Page 10, line 255: you briefly mentioned the role of abnormal activity in various neurological disorders. Recently, a study by Deng et al. (PMID: 33679331) evaluated the application of IDO1 and different kynurenine pathway metabolites in treatment of epilepsy. This reference would be a useful asset to your manuscript.
Response: We thank the reviewer for this valuable suggestion. We have now incorporated this reference in the revised manuscript (Line 275-279, Line 401-402).

Reviewer 3 Report
The authors described changes in cerebral levels of Trp, KYN and KYNA after pilocarpine administration in rodents. In addition, the electrophysiological effect of extracellular KYNA on brain sections in pilocarpine-treated animals was evaluated. Although this is an interesting study, several points need to be addressed in the manuscript.
1. The figures must be represented by individual points, in this way the effect will be better appreciated, since, as shown, the dispersions are very large between the groups.
2. The number of animals used in each determination should be mentioned.
3. Each figure legend should indicate what the bars represent, along with the n and statistics used.
4. The authors mention that they measure the activity of KAT with the KYN/KYNA ratio, however, in tissue this ratio is not indicative of the enzyme's own activity, since it has been described that KYNA can be produced by other pathways favored by the redox state. In this context, it is important to describe whether the model addressed in this work induces the formation of reactive oxygen species that could be contributing to the formation of KYNA through a non-canonical pathway. I suggest leaving only the Kyn/KYNA ratio without mentioning that it is a measure of KAT activity.
5. It should be discussed why the PLP levels do not change. If the authors attribute a decrease in KAT activity, would PLP levels not increase?
6. It should be discussed whether the increase in KYN leads to an increase in the metabolites of the long arm of KP, which can also exacerbate the effect of pilocarpine.
7. Would the use of diazepam to control seizures have any effect on KP catabolism? Were controls made in this regard?
8. In the methods, it is necessary to add information such as the retention time for each KP metabolite measured. Was the same phase used for the determination of KP metabolites?
Author Response
Comment: The authors described changes in cerebral levels of Trp, KYN and KYNA after pilocarpine administration in rodents. In addition, the electrophysiological effect of extracellular KYNA on brain sections in pilocarpine-treated animals was evaluated. Although this is an interesting study, several points need to be addressed in the manuscript.
Response: We thank the reviewer for going through our manuscript and providing valuable comments on our manuscript. We have now provided responses to all the queries raised by the reviewer and most of them has been incorporated in the revised manuscript.
Comment: The figures must be represented by individual points, in this way the effect will be better appreciated, since, as shown, the dispersions are very large between the groups.
Response: We thank the reviewer for this suggestion. We have now represented the figures by individual points.
Comment: The number of animals used in each determination should be mentioned.
Response: We have now mentioned the animal number in revised figure legends 3 and 4 (Line 200-201, Line 246).
Comment: Each figure legend should indicate what the bars represent, along with the n and statistics used.
Response: We have included the same in the revised manuscript (Line 200, 219, 245).
Comment: The authors mention that they measure the activity of KAT with the KYN/KYNA ratio, however, in tissue this ratio is not indicative of the enzyme's own activity, since it has been described that KYNA can be produced by other pathways favored by the redox state. In this context, it is important to describe whether the model addressed in this work induces the formation of reactive oxygen species that could be contributing to the formation of KYNA through a non-canonical pathway. I suggest leaving only the Kyn/KYNA ratio without mentioning that it is a measure of KAT activity.
Response: We thank the reviewer for pointing out this fact that, besides KAT II, KYNA can be synthesized from D-kynurenine(D-KYN) through oxidative deamination by D-amino acid oxidase (DAAO) and spontaneous conversion in presence of reactive oxygen species (ROS). However, KAT II mainly acts on L-KYN to synthesize KYNA. Here, we found a reduction in the level of KYNA following status epilepticus. So, it is logical to assume that alternative routes for KYNA synthesis have not contributed but rather due to dysfunction of KAT II, the level of KYNA was reduced in the hippocampal samples.
Instead of substrate product ratio (TRP/KYN and KYN/KYNA), we have now calculated it to product substrate ratio (KYN/TRP and KYNA/KYN) which reflects KYNA activity more accurately and included in the revised figure 3 as well as in the manuscript.
Comment: It should be discussed why the PLP levels do not change. If the authors attribute a decrease in KAT activity, would PLP levels not increase?
Response: In the present study we have used acute model of TLE where we could not see altered PLP levels. Experiments in animal models with chronic period after pilocarpine administration will help to assess the change in the level of PLP more accurately. We have now mentioned this in the discussion section of the revised manuscript (Line 280-282).
Comment: It should be discussed whether the increase in KYN leads to an increase in the metabolites of the long arm of KP, which can also exacerbate the effect of pilocarpine.
Response: We thank the reviewer for this suggestion. We have now mentioned this in the discussion section of the manuscript (Line 271-279).
Comment: Would the use of diazepam to control seizures have any effect on KP catabolism? Were controls made in this regard?
Response: We could not check the effect of diazepam on KP metabolism in the present study. This is a limitation of the present study and we have added this to the discussion section of the manuscript (Line 310-312)
Comment: In the methods, it is necessary to add information such as the retention time for each KP metabolite measured. Was the same phase used for the determination of KP metabolites?
Response: We have now added the retention time of the metabolites in the methods section (Line 114-115). The mobile phase was same for all the metabolites.

Round 2
Reviewer 1 Report
minor points:
1. Fig. 3, Legend - correct (D) and (E)
2. Correct Table 1 - there is still: "Single digit denotes p<0.05, double digits denote p<0.01, triple digits denote p<0.001".
3. Check the entire text carefully because there are still errors, such as: "The ratio of tryptophan-kynurenine" (line 189); "The ratio of kynurenine-kynurenic acid (lines 191-192).
Author Response
Comment: Fig. 3, Legend - correct (D) and (E)
Response: We have now corrected the legend of Figure 3 in the revised manuscript (Line 198).
Comment: Correct Table 1 - there is still: "Single digit denotes p<0.05, double digits denote p<0.01, triple digits denote p<0.001".
Response: We have now mentioned only p<0.05 in the legend of table 1 (Line 219).
Comment: Check the entire text carefully because there are still errors, such as: "The ratio of tryptophan-kynurenine" (line 189); "The ratio of kynurenine-kynurenic acid (lines 191-192).
Response: We thank the reviewer for pointing out these errors. We have now corrected the manuscript (Line 189, 191-192).